# Localized statistics decoding for quantum low-density parity-check codes

Timo Hillmann [1,7] ✉, Lucas Berent [2,7] ✉, Armanda O. Quintavalle[3], Jens Eisert [3,4], Robert Wille[2,5] & Joschka Roffe [3,6] ✉

Quantum low-density parity-check codes are a promising candidate for fault-tolerant quantum computing with considerably reduced overhead compared to the surface code. However, the lack of a practical decoding algorithm remains a barrier to their implementation. In this work, we introduce localized statistics decoding, a reliability-guided inversion decoder that is highly parallelizable and applicable to arbitrary quantum low-density parity-check codes. Our approach employs a parallel matrix factorization strategy, which we call *on-the-fly elimination*, to identify, validate, and solve local decoding regions on the decoding graph. Through numerical simulations, we show that localized statistics decoding matches the performance of state-of-the-art decoders while reducing the runtime complexity for operation in the sub-threshold regime. Importantly, our decoder is more amenable to implementation on specialized hardware, positioning it as a promising candidate for decoding real-time syndromes from experiments.

*Quantum low-density parity-check* (QLDPC) codes[1] are a promising alternative to the surface code[2–4]. Based on established methods underpinning classical technologies such as Ethernet and 5G[5,6], QLDPC codes promise a low-overhead route to fault tolerance[7–13], encoding multiple qubits per logical block as opposed to a single one for the surface code. While, as a trade-off, QLDPC codes require long-range interactions that can be difficult to implement physically, various architectures allow for those requirements[14–17]. In particular, recent work targeting quantum processors based on neutral atom arrays[13] as well a bi-layer superconducting qubit chip architecture[12] suggest that QLDPC codes can achieve an order-of-magnitude reduction in overhead relative to the surface code on near-term hardware.

In a quantum error correction circuit, errors are detected by measuring stabilizers yielding a stream of syndrome information. The *decoder* is the classical co-processor tasked with performing real-time inference on the measured error syndromes to determine a correction operation that must take place within a time frame less than the decoherence time of the physical qubits. Full-scale quantum computers will impose significant demands on their decoders, with estimates suggesting that terabytes of decoding bandwidth will be required for real-time processing of syndrome data[18,19]. As such, decoding algorithms must be as efficient as possible and, in particular, suitable for parallel implementation on specialized hardware[20].

The current gold standard for decoding general QLDPC codes is the *belief propagation plus ordered statistics decoder* (BP+OSD)[11,21]. The core of this decoder is the iterative *belief propagation* (BP) algorithm[22] that finds widespread application in classical error correction. Unfortunately, BP decoders are not effective out of the box for QLDPC codes. The reason for this shortcoming are so-called *degenerate errors*, that is, physically different errors that are equivalent up to stabilizers and prevent BP from converging[11,23,24]. The BP+OSD algorithm augments BP with a post-processing routine based on *ordered statistics decoding* (OSD)[11,21,25,26]. OSD is invoked if the BP algorithm fails to converge and computes a solution by inverting a full-rank submatrix of the parity check matrix. A specific strength of the BP+OSD decoder lies in its versatility: it achieves good decoding performance across the landscape of quantum LDPC codes[21].

[1]Chalmers University of Technology, Gothenburg, Sweden. [2]Technical University of Munich, Munich, Germany. [3]Freie Universität Berlin, Berlin, Germany. [4]Helmholtz-Zentrum Berlin für Materialien und Energie, Berlin, Germany. [5]Software Competence Center Hagenberg, Hagenberg, Austria. [6]University of Edinburgh, Edinburgh, United Kingdom. [7]These authors contributed equally: Timo Hillmann, Lucas Berent. ✉e-mail: timo.hillmann@rwth-aachen.de; lucas.berent@tum.de; joschka@roffe.eu

A significant limitation of the BP+OSD decoder is its large runtime overhead. This inefficiency stems primarily from the OSD algorithm's inversion step, which relies on Gaussian elimination and has cubic worst-case time complexity in the size of the corresponding check matrix. In practice, this is a particularly acute problem, as decoders must be run on large circuit-level decoding graphs that account for errors occurring at any location in the syndrome extraction circuit. This shortcoming constitutes a known barrier to the experimental implementation of efficient quantum codes, as circuit-level decoding graphs can contain tens of thousands of nodes[12]. Even with specialized hardware, inverting the matrix of a graph of this size cannot realistically be achieved within the decoherence time of a typical qubit[27]. Whilst the BP+OSD decoder is a useful tool for simulations, it is not generally considered a practical method for real-time decoding.

In this work, we introduce *localized statistics decoding* (LSD) as a parallel and efficient decoder for QLDPC codes, designed specifically to address the aforementioned limitations of BP+OSD, while retaining generality and good decoding performance. The key idea underpinning LSD is that in the sub-threshold regime, errors typically span disconnected areas of the decoding graph. Instead of inverting the entire decoding graph, LSD applies matrix inversion independently and concurrently for the individual *sub-graphs* associated with these decoding regions. Similar to OSD, the performance of LSD can be improved using the soft information output of a pre-decoder such as BP. Our numerical decoding simulations of surface codes, bicycle bivariate codes, and hypergraph product codes show that our implementation of the BP+LSD decoder performs on par with BP+OSD in terms of decoding performance.

The efficiency of the LSD algorithm is made possible by a new linear algebra routine, which we call *on-the-fly elimination*, that transforms the serial process of Gaussian elimination into a parallel one. Specifically, our method allows separate regions of the decoding graph to be reduced on separate processors. A distinct feature of on-the-fly elimination lies in a sub-routine that efficiently manages the extension and merging of decoding regions without necessitating the re-computation of row operations. The methods we introduce promise reduced runtime in the sub-threshold regime and open the possibility of using inversion-based decoders to decode real syndrome information from quantum computing experiments. We anticipate that on-the-fly elimination will also find broader utility in efficiently solving sparse linear systems across various settings, such as recommender systems[28] or compressed sensing[29].

## Results

### The decoding problem

In this paper, we focus on the *Calderbank-Shor-Steane* (CSS) subclass of QLDPC codes. These codes are defined by constant weight Pauli-*X* and -*Z* operators called *checks* that generate the stabilizer group defining the code space. In a gate-based model of computation, the checks are measured using a circuit containing auxiliary qubits and two-qubit Clifford gates that map the expectation value of each check onto the state of an auxiliary qubit. The circuit that implements all check measurements is called the *syndrome extraction circuit*.

For the decoding of QLDPC codes, the decoder is provided with a matrix $H \in \mathbb{F}^{|D| \times |F|}$ called the *detector check matrix*. This matrix maps circuit fault locations $F$ to so-called *detectors* $D$, defined as linear combinations of check measurement outcomes that are deterministic in the absence of errors. Specifically, each row of $H$ corresponds to a detector and each column to a fault, and $H_{df} = 1$ if fault $f \in \{1, ..., |F|\}$ flips detector $d \in \{1, ..., |D|\}$. Such a check matrix can be constructed by tracking the propagation of errors through the syndrome extraction circuit using a stabilizer simulator[30–32].

We emphasize that, once the detector matrix $H$ is created, the minium-weight decoding problem can be mapped to the problem of decoding a classical linear code: Given a *syndrome* $\mathbf{s} \in \mathbb{F}_2^{|D|}$, the

*decoding problem* consists of finding a minimum-weight recovery $\hat{\mathbf{e}}$ such that $\mathbf{s} = H \cdot \hat{\mathbf{e}}$, where the vector $\hat{\mathbf{e}} \in \mathbb{F}_2^{|F|}$ indicates the locations in the circuit where faults have occurred.

The *decoding graph* is a bipartite graph $\mathcal{G}(H) = (V_D \cup V_F, E)$ with *detector* nodes $V_D$, *fault* nodes $V_F$ and edges $(d, f) \in E \Leftrightarrow H_{df} = 1$. $\mathcal{G}(H)$ is also known as the Tanner graph of the check matrix $H$. Since the detector check matrix is analogous to a parity check matrix of a classical linear code, we use the terms detectors and checks synonymously. Note that we implement a minimum-weight decoding strategy where the goal is to find the lowest-weight error compatible with the syndrome. This is distinct from maximum-likelihood decoding, where the goal is to determine the highest probability logical coset.

### Localized statistics decoding

This section provides an example-guided outline of the localized statistics decoder. A more formal treatment, including pseudo-code, can be found in Methods.

*a. Notation.* For an index set $I = \{i_1, ..., i_n\}$ and a matrix $M = (m_1, ..., m_\ell)$ with columns $m_j$, we write $M_{[I]} = (m_{i_1}, ..., m_{i_n})$ as the matrix containing only the columns indexed by $I$. Equivalently, for a vector $\mathbf{v}$, $\mathbf{v}_{[I]}$ is the vector containing only coordinates indexed by $I$.

*b. Inversion decoding.* The *localized statistics decoding* (LSD) algorithm belongs to the class of reliability-guided inversion decoders, which also contains *ordered statistics decoding* (OSD)[11,21,26]. OSD can solve the decoding problem by computing $\hat{\mathbf{e}}_{[I]} = H_{[I]}^{-1} \cdot \mathbf{s}$. Here, $H_{[I]}$ is an invertible matrix formed by selecting a linearly independent subset of the columns of the check matrix $H$ indexed by the set of column indices $I$. The algorithm is reliability-guided in that it uses prior knowledge of the error distribution to strategically select $I$ so that the solution $\hat{\mathbf{e}}_{[I]}$ spans faults that have the highest error probability. The reliabilities can be derived, for example, from the device's physical error model[16,33–37] or the soft information output of a *pre-decoder* such as BP[38].

*c. Factorizing the decoding problem.* In general, solving the system $\hat{\mathbf{e}}_{[I]} = H_{[I]}^{-1} \cdot \mathbf{s}$ involves applying Gaussian elimination to compute the inverse $H_{[I]}^{-1}$, which has cubic worst-case time complexity, $O(n^3)$, in the size $n$ of the check matrix $H$. The essential idea behind the LSD decoder is that, for low physical error rates, the decoding problem for QLDPC amounts to solving a sparse system of linear equations. In this setting, the inversion decoding problem can be factorized into a set of independent linear sub-systems that can be solved concurrently.

Figure 1 shows an example of error factorization in the Tanner graph of a $5 \times 10$ surface code. The support of a fault vector $\mathbf{e}$ is illustrated by the circular nodes marked with an $X$ and the corresponding syndrome is depicted by the square nodes filled in red. In this example, it is clear that $\mathbf{e}$ can be split into two connected components, $\mathbf{e}_{[C_1]}$ and $\mathbf{e}_{[C_2]}$, that occupy separate regions of the decoding graph. We

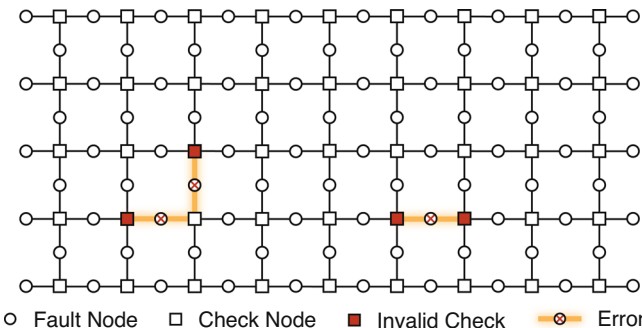

○ Fault Node   □ Check Node   ■ Invalid Check   ⊗— Error

**Fig. 1 | Illustration of the factorization of the decoding problem on a 5 × 10 surface code patch.** Below the threshold, errors are typically sparsely distributed on the decoding graph and form small clusters with disjoint support.

refer to each of the connected components induced by an error on the decoding graph as *clusters*. With a slight misuse of notation, we refer to clusters $C_i$ and their associated incidence matrices $H_{[C_i]}$ interchangeably and use $|C_i|$ to denote the number of fault nodes (columns) in the cluster (its incidence matrix, respectively). This identification is natural as clusters are uniquely identified by their fault nodes, or equivalently, by column indices of $H$: for a set of fault nodes $C \subseteq V_F$, we consider all of the detector nodes in $V_D$ adjacent to at least one node in $C$ to be part of the cluster.

For the example in Fig. 1, the two induced clusters $H_{[C_1]}, H_{[C_2]}$ are entirely independent of one another. As such, it is possible to find a decoding solution by inverting each submatrix separately,

$$\hat{\mathbf{e}}_{[C_1 \cup C_2 \cup C_\perp]} = \left( H_{[C_1]}^{-1} \mathbf{s}_{[C_1]}, H_{[C_2]}^{-1} \mathbf{s}_{[C_2]}, 0 \right), \qquad (1)$$

where $\mathbf{s}_{[C_i]}$ is the subset of syndrome bits in the cluster $H_{[C_i]}$, which we refer to as the *cluster syndrome*. The set $C_\perp$ is the column index set of fault nodes that are not in any cluster.

In general, linear systems can be factorized into $\nu$ many decoupled clusters, yielding

$$\hat{\mathbf{e}}_{[C_1 \cup \ldots \cup C_\nu \cup C_\perp]} = \left( H_{[C_1]}^{-1} \mathbf{s}_{[C_1]}, \ldots, H_{[C_\nu]}^{-1} \mathbf{s}_{[C_\nu]}, 0 \right). \qquad (2)$$

The number of clusters, $\nu$, will depend upon $H$, the physical error rate, and the Hamming weight of $\mathbf{s}$. If a factorization can be found, matrix inversion is efficient: first, the $\nu$ clusters can be solved in parallel; second, the parallel worst-case time complexity of the algorithm depends on the maximum size of a cluster $\kappa = \max_i(|C_i|)$, where $|C_i|$ is the number of fault nodes in $C_i$. The worst-case scaling $O(\kappa^3)$ contrasts with the $O(n^3)$ OSD post-processing scaling, where $n = |V_F|$ is the size of the matrix $H$. To enable parallel execution, we have devised a routine that we call *on-the-fly elimination* to efficiently merge clusters and compute a matrix factorization, as detailed in the Methods.

*d. Weighted cluster growth and the LSD validity condition.* For a given syndrome, the LSD algorithm is designed to find a factorization of the decoding graph that is as close to the optimal factorization as possible. Here, we define a factorization as *optimal* if its clusters correspond exactly to the connected components induced by the error.

The LSD decoder uses a weighted, reliability-based growth strategy to factorize the decoding graph. The algorithm begins by creating a cluster $H_{[C_i]}$ for each flipped detector node, i.e., a separate cluster is generated for every nonzero bit in the syndrome vector $\mathbf{s}$. At every growth step, each cluster $H_{[C_i]}$ is grown by one column by adding the fault node from its neighborhood with the highest probability of being in error according to the input reliability information. This weighted growth strategy is crucial for controlling the cluster size: limiting growth to a single fault node per time step increases the likelihood that

an efficient factorization is found, especially for QLDPC codes with high degrees of expansion in their decoding graphs.

If two or more clusters collide – that is, if a check node would be contained in multiple clusters after a growth step – the LSD algorithm merges them and forms a combined cluster. We use the notation $H_{[C_1 \cup C_2]}$ and $\mathbf{s}_{[C_1 \cup C_2]}$ to indicate the decoding matrix and the syndrome of the combined cluster.

For each cluster $C_i$, the LSD algorithm iterates cluster growth until it has enough linearly independent columns to find a local solution, i.e., until $\mathbf{s}_{[C_i]} \in \text{image}(H_{[C_i]})$. We call such a cluster *valid*. Once all clusters are valid, the LSD algorithm computes all local solutions, $\mathbf{e}_{[C_i]} = H_{[C_i]}^{-1} \cdot \mathbf{s}_{[C_i]}$, and combines them into a global one.

The process of weighted-cluster growth is conceptually similar to "belief hypergraph union-find"[39] and is illustrated in Fig. 2 for the surface code. Here, two clusters are created. These are grown according to the reliability ordering of the neighboring fault nodes. In Fig. 2c, the two clusters merge, yielding a combined valid cluster. The combined cluster is not optimal as its associated decoding matrix has 5 columns, whereas the local solution has Hamming weight 3, indicating that the optimal cluster would have 3 columns. Nonetheless, computing a solution using the cluster matrix with 5 columns is still preferable to computing a solution using the full 41-column decoding matrix – this highlights the possible computational gain of LSD.

*e. On-the-fly elimination and parallel implementation.* To avoid the overhead incurred by checking the validity condition after each growth step – a bottleneck for other clustering decoders for QLDPC codes[40] – we have developed an efficient algorithm that we call *on-the-fly elimination*. Our algorithm maintains a dedicated data structure that allows for efficient computation of a matrix factorization of each cluster when additional columns are added to the cluster, even if clusters merge – see Methods for details. Importantly, at each growth step, due to our on-the-fly technique, we only need to eliminate a single additional column vector without having to re-eliminate columns from previous growth steps.

Crucially, on-the-fly elimination can be applied in parallel to each cluster $H_{[C_i]}$. Using the on-the-fly data structure that enables clusters to be efficiently extended without having to recompute their new factorization from scratch, we propose a fully *parallel implementation of LSD* in Section 2 of the Supplementary Material. There, we analyze parallel LSD time complexity and show that the overhead for each parallel resource is low and predominantly depends on the cluster sizes.

*f. Factorization in decoding graphs.* A key feature of LSD is to divide the decoding problem into smaller, local sub-problems that correspond to error clusters on the decoding graph. To provide more insight, we investigate cluster formation under a specific noise model and compare these clusters obtained directly from the error to the clusters identified by LSD.

**(a)**  **(b)**  **(c)**  **(d)**

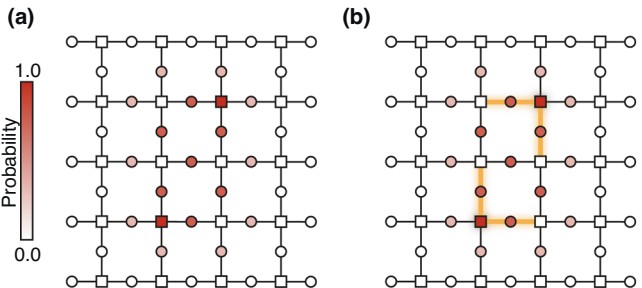
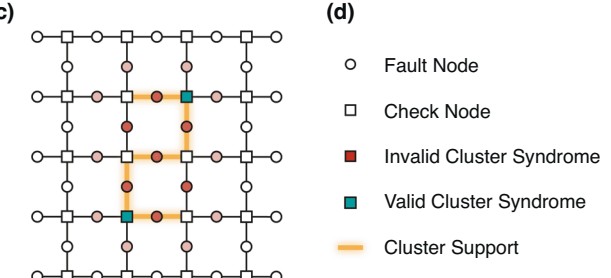

**Fig. 2 | Reliability-based weighted cluster growth example for the surface code.** **a** The syndrome of an error is indicated as red square vertices. The fault nodes are colored to visualize their error probabilities obtained from belief propagation preprocessing. **b** Clusters after the first two growth steps. In the guided cluster growth strategy, fault nodes are added individually to the local clusters. The order of adding the first two fault nodes to each cluster is random since both have the same probability due to the presence of degenerate errors. **c** After an additional growth step, the two clusters are merged and the combined cluster is valid. **d** Legend for the used symbols.

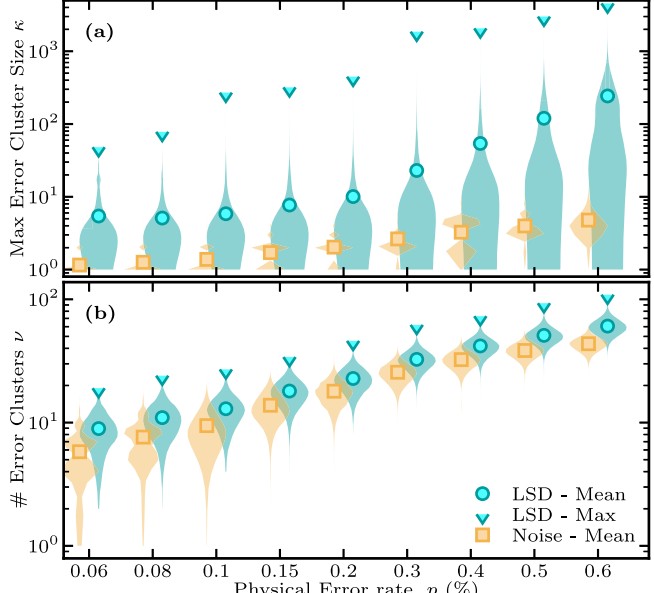

**Fig. 3 | Cluster size statistics of the [[144, 12, 12]] bivariate bicycle code of ref. 12 under circuit-level noise with strength *p*.** Markers show the mean of the distribution while shapes are violin plots of the distribution obtained from $10^5$ samples. Yellow distributions show statistics for the optimal factorization while the blue distributions show statistics for the factorization returned by BP+LSD. We show in (**a**) the distribution of the maximum cluster size $\kappa$ and in (**b**) the distribution of the cluster count, $\nu$, for each decoding sample. Markers and distributions are slightly offset from the actual error rate to increase readability.

As a timely example, we focus specifically on the cluster size statistics of the circuit-noise decoding graph of the [[144,12,12]] bivariate bicycle code[41] that was recently investigated in ref. 12. Figure 3a shows the distribution of the maximum sizes of clusters identified by BP+LSD over $10^5$ decoding samples, see Methods for details. The figure illustrates that for low enough noise rates, the largest clusters found by LSD are small and close to the optimal sizes of clusters induced by the original error, even if only a relatively small number (30) of BP iterations is used to compute the soft information input to LSD. It is worth emphasizing that large clusters are typically formed by merging two, or more, smaller clusters identified and processed at previous iterations of the algorithm. Owing to our on-the-fly technique that processes the linear system corresponding to each of these clusters (cf. Methods), the maximum cluster size only represents a loose upper bound on the complexity of the LSD algorithm.

Figure 3 b shows the distributions of the cluster count per shot, $\nu$ – that is, per shot, where the LSD data is post-selected on shots where BP does not converge – against the physical error rate $p$. The number of clusters $\nu$ corresponds to the number of terms in the factorization of the decoding problem and thus indicates the degree to which the decoding can be parallelized, as disjoint factors can be solved concurrently. At practically relevant error rates below the (pseudo) threshold, e.g., $p \leq 0.1\%$, we observe on average 10 independent clusters. This implies that the LSD algorithm benefits from parallel resources throughout its execution.

We explore bounds on the sizes of clusters induced by errors on QLDPC code graphs in Section 1 of the Supplementary Material. Our findings suggest that detector matrices generally exhibit a strong suitability for factorization, a feature that the LSD algorithm is designed to capitalize on.

*g. Higher-order reprocessing.* Higher-order reprocessing in OSD is a systematic approach designed to increase the decoder's accuracy. The zero-order solution $\hat{\mathbf{e}}_{[I]} = H_{[I]}^{-1} \cdot \mathbf{s}$ of the decoder cannot be made lower

weight if the set of column indices $I$ specifying the invertible submatrix $H_{[I]}$ matches the $|I|$ most likely fault locations identified from the soft information vector $\boldsymbol{\lambda}$. However, if there are linear dependencies within the columns formed by the $|I|$ most likely fault locations, the solution $\hat{\mathbf{e}}$ may not be optimal. In those cases, some fault locations in $\bar{I}$ (the complement of $I$) might have higher error probabilities. To find the optimal solution, one can systematically search all valid fault configurations in $\bar{I}$ that potentially provide a more likely estimate $\hat{\mathbf{e}}\prime$. This search space, however, is exponentially large in $|\bar{I}|$. Thus, in practice, only configurations with a Hamming weight up to $w$ are considered, known as *order-w* reprocessing. See refs. 11,21,25,26 for a more technical discussion.

In BP+OSD-*w* applied to $H$, order-*w* reprocessing is frequently the computational bottleneck because of the extensive search space and the necessary matrix-vector multiplications involving $H_{[I]}$ and $H_{[\bar{I}]}$ to validate fault configurations. Inspired by higher-order OSD, we propose a higher-order reprocessing method for LSD, which we refer to as LSD − $\mu$. We find that when higher-order reprocessing is applied to LSD, it is sufficient to process clusters locally. This offers three key advantages: parallel reprocessing, a reduced higher-order search space, and smaller matrix-vector multiplications. Furthermore, our numerical simulations indicate that decoding improvements of local BP+LSD − $\mu$ are on par with those of global BP+OSD − $w$. For more details on higher-order reprocessing with LSD and additional numerical results, see Section 4 of the Supplementary Material.

## Numerical results

For the numerical simulations in this work, we implement serial LSD, where the reliability information is provided by a BP pre-decoder. The BP decoder is run in the first instance, and if no solution is found, LSD is invoked as a post-processor. Our serial implementation of this BP+LSD decoder is written in C++ with a python interface and is available open-source as part of the `LDPC` package[42].

Our main numerical finding is that BP+LSD can decode QLDPC codes with performance on par with BP+OSD. We include the results of extensive simulations in which BP+LSD is used to decode a circuit-level depolarizing noise model for surface codes, hypergraph product (HGP) codes[43], and bivariate bicycle codes[12,41].

In BP+OSD decoding, it is common to run many BP iterations to maximize the chance of convergence and reduce the reliance on OSD post-processing. A strength of the BP+LSD decoder is that LSD is less costly than OSD and, therefore, applying the LSD routine after running BP introduces comparatively small overall computational overhead. As a result, the number of BP iterations in BP+LSD can be considerably reduced since LSD requires only a few BP iterations to obtain meaningful soft information values. This is in stark contrast to BP+OSD, where it is often more efficient to run many BP iterations rather than deferring to costly OSD. In this work, we use a fixed number of 30 BP iterations for all decoding simulations with BP+LSD. For context, this is a significant reduction compared to the decoding simulations of ref. 12 where BP+OSD was run with $10^4$ BP iterations.

*a. Surface codes.* We compare the threshold of BP+LSD with various state-of-the-art decoders that are similarly guided by the soft information output of a BP decoder. In particular, we compare the proposed BP+LSD algorithm with BP+OSD (order 0)[21], as well as our implementation of a BP plus union-find (BP+UF) decoder[44] that is tailored to matchable codes. The results are shown in Fig. 4. The main result is that both BP+OSD and BP+LSD achieve a similar threshold close to a physical error rate of $p \approx 0.7\%$, and similar logical error rates, see panels (a) and (c), respectively. In particular, in the relevant sub-threshold regime, where BP+LSD can be run in parallel, its logical decoding performance matches BP+OSD. Note that this is the desired outcome and demonstrates that our algorithm achieves (close to) identical performance with BP+OSD while maintaining locality. Our implementation of the BP+UF decoder of ref. 38, see panel (b),

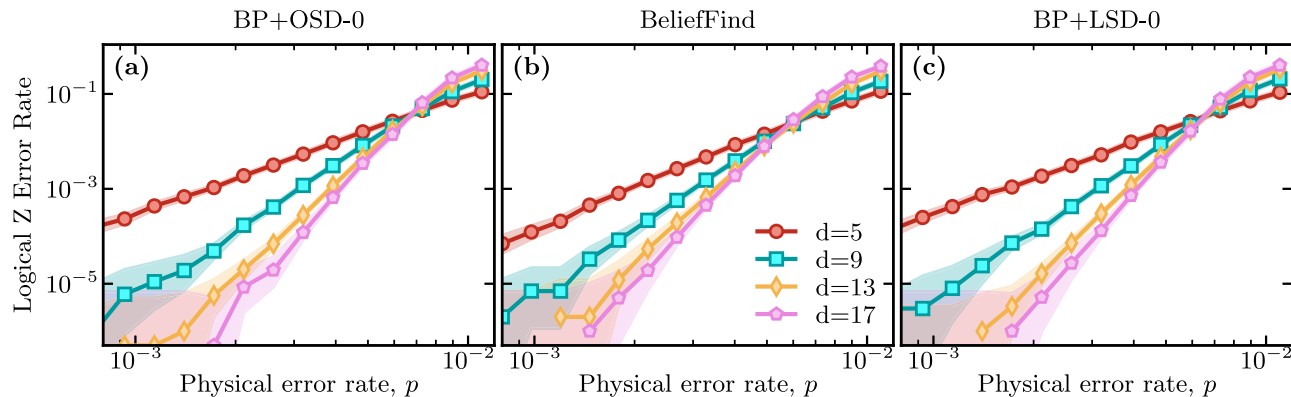

**Fig. 4 | Comparison of various decoders guided by belief propagation for decoding rotated surface codes of distance *d* subject to circuit-level depolarizing noise parameterized by a single parameter, called the *physical error rate p*, see Section IV C for details.** We use `Stim` to perform a `surface_code:rotated_memory_z` experiment for *d* syndrome extraction cycles with single and two-qubit error probabilities *p*. **a** The performance of BP+OSD-0 that matches the

performance of the proposed decoder. **b** The performance of a BeliefFind decoder that shares a cluster growth strategy with the proposed decoder. **c** Performance of the proposed BP+LSD decoder. The shading indicates hypotheses whose likelihoods are within a factor of 1000 of the maximum likelihood estimate, similar to a confidence interval.

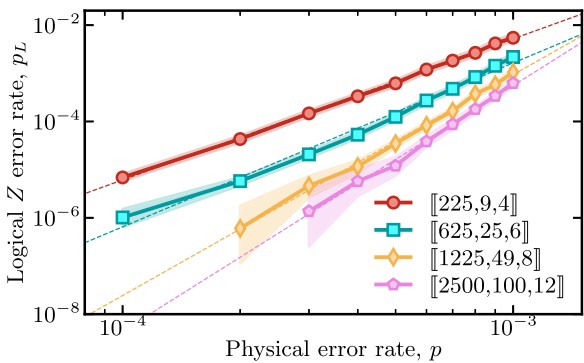

**Fig. 5 | Below threshold logical error rate $p_L$ of a family of [[$25s^2$, $s^2$]] constant-rate hypergraph product codes decoded with the BP+LSD decoder.** We simulate $N_c = 12$ rounds of syndrome extraction cycles under circuit-level noise with physical error rate *p* and apply a (3, 1)-overlapping window technique to enable fast and accurate single-shot decoding, see Methods for details. The shading indicates hypotheses whose likelihoods are within a factor of 1000 of the maximum likelihood estimate, similar to a confidence interval. Dashed lines are an exponential fit with a linear exponent to the numerically observed error rates.

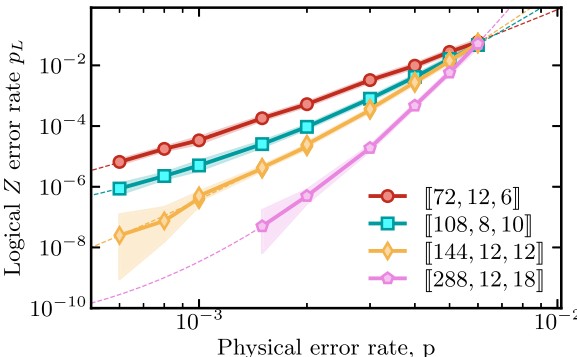

**Fig. 6 | Logical error rate per syndrome cycle $p_L$ for various bivariate bicycle codes under a circuit-level noise model.** For each code, *d* rounds of syndrome extraction are simulated and the full syndrome history is decoded using BP+LSD. The shading highlights the region of estimated probabilities where the likelihood ratio is within a factor of 1000, similar to a confidence interval. Dashed lines are an exponential fit with a quadratic exponent to the numerically observed error rates.

performs slightly worse, achieving a threshold closer to $p \approx 0.6\%$ and higher logical error rates, potentially due to a non-optimized implementation.

*b. Random (3,4)-regular hypergraph product codes.* Fig. 5 shows the results of decoding simulations for the family of hypergraph product codes[43] that were recently studied in ref. 13. The plot shows the logical error rate per syndrome cycle $p_L = 1 - (1 - P_L(N_c))^{1/N_c}$, where $N_c$ is the number of syndrome cycles and $P_L(N_c)$ the logical error rate after $N_c$ rounds. Under the assumption of an identical, independent circuit-level noise model, BP+LSD significantly outperforms the BP plus small set-flip (BP+SSF) decoder investigated in ref. 45. For example, for the [[625, 25]] code instance at $p \approx 0.1\%$, BP+LSD improves logical error suppression by almost two orders of magnitude compared to BP+SSF.

*c. Bivariate bicycle codes.* Here, we present decoding simulation results of the *bivariate bicycle* (BB) codes. These codes are part of the family of hyperbicycle codes originally introduced in ref. 41, and more recently investigated at the circuit level in ref. 12. In Fig. 6, we show the logical error *Z* rate per syndrome cycle, $p_{L_z}$. We find that with BP+LSD we obtain comparable decoding performance to the results presented in ref. 12 where simulations were run using BP+OSD-CS-7 (where BP

+OSD-CS-7 refers to the "combination sweep" strategy for BP+OSD higher-order processing with order $w = 7$, see ref. 21 for more details).

*d. Runtime statistics.* To estimate the time overhead of the proposed decoder in numerical simulation scenarios and to compare it with a state-of-the-art implementation of BP+OSD, we present preliminary timing results for our prototypical open-source implementation of LSD in Section 3.4 of the Supplementary Material. We note that for a more complete assessment of performance, it will be necessary to benchmark a fully parallel implementation of the algorithm, designed for specialized hardware such as GPUs or FPGAs. We leave this as a topic for future work.

## Discussion
When considering large QLDPC codes, current state-of-the-art decoders such as BP+OSD hit fundamental limitations due to the size of the resulting decoding graphs. This limitation constitutes a severe bottleneck in the realization of protocols based on QLDPC codes. In this work, we address this challenge through the introduction of the LSD decoder as a parallel algorithm whose runtime depends predominantly on the physical error rate of the system. Our algorithm uses a reliability-based growth procedure to construct clusters on the

decoding graph in a parallel fashion. Using a novel routine that computes the PLU decomposition[46] of the clusters' sub-matrices on-the-fly, we can merge clusters efficiently and compute local decoding solutions in a parallel fashion. Our main numerical findings are that the proposed decoder performs on par with current state-of-the-art decoding methods in terms of logical decoding performance.

A practical implementation of the algorithm has to be runtime efficient enough to overcome the so-called backlog problem[47], where syndrome data accumulates since the decoder is not fast enough. While we have implemented an overlapping window decoding technique for our algorithm, it might be interesting to further investigate the performance of LSD under parallel window decoding[20], where the overlapping decoding window is subdivided to allow for further parallelization of syndrome data decoding.

To decode syndrome data from quantum computing experiments in real-time, it will be necessary to use specialized hardware such as *field programmable gate arrays* (FPGAs) or *application-specific integrated circuits* (ASICs), as recently demonstrated for variants of the union-find surface code decoder[48–50] or possibly cellular automaton based approaches[51]. A promising avenue for future research is to explore the implementation of an LSD decoder on such hardware to assess its performance with real-time syndrome measurements.

Concerning alternative noise models, erasure-biased systems have recently been widely investigated[52–55]. We conjecture that LSD can readily be generalized to erasure decoding, either by adapting the cluster initialization or by considering a re-weighting procedure of the input reliabilities. We leave a numerical analysis as a topic for future work.

Finally, it would be interesting to investigate the use of maximum-likelihood decoding at the cluster level as recently explored in ref. 56 as part of the BP plus ambiguity clustering (BP+AC) decoder. Specifically, such a method could improve the efficiency of the LSD – $\mu$ higher-order reprocessing routines we explored. Similarly, the BP+AC decoder could benefit from the results of this paper: our parallel LSD cluster growth strategy, combined with on-the-fly elimination, provides an efficient strategy for finding the BP+AC block structure using parallel hardware.

## Methods
### LSD algorithm
In this section, we provide a detailed description of the LSD algorithm and its underlying data structure designed for efficient cluster growth, merging, validation, and ultimately local inversion decoding. We start with some foundational definitions.

**Definition 1.1.** (Clusters). Let $\mathcal{G}(H) = (V_D \cup V_F, E)$ be the decoding graph of a QLDPC code with detector nodes $V_D$ and fault nodes $V_F$. There exists an edge $(d, f) \in E \Leftrightarrow H_{df} = 1$. A *cluster* $C = (V_D^C \cup V_F^C, E^C) \subseteq \mathcal{G}(H)$ is a connected component of the decoding graph.

**Definition 1.2.** (Cluster sub-matrix). Given a set of column indices $C$ of a cluster, the sub-matrix $H_{[C]}$ of the check matrix $H$ is called the *cluster sub-matrix*. The *local syndrome* $s_{[C]}$ of a cluster is the support vector of detector nodes in the cluster. A cluster is *valid* if $s_{[C]} \in \text{IMAGE}(H_{[C]})$. Note that a cluster is uniquely identified by the columns of its sub-matrix $H_{[C]}$, hence we use $H_{[C]}$ to denote both the cluster and its sub-matrix.

**Definition 1.3.** (Cluster-boundary and candidate fault nodes) The set of *boundary detector* nodes $\beta(C) \subseteq V_D^C$ of a cluster $C$ is the set

$$\beta(C) = \{d \in V_D^C | \Gamma(d) \nsubseteq \Gamma_F^C\} \tag{3}$$

of all detector nodes in $C$ that are connected to at least one fault node not in $C$, where $\Lambda(v)$ is the neighborhood of the vertex v, i.e.,

$\Lambda(v) = \{u \in \mathcal{G}(H) | (v, u) \in E\}$. We define *candidate fault nodes* $\Lambda(C) \subseteq V_F \setminus V_F^C$ as the set of fault nodes not in $C$ and connected to at least one boundary detector node in $\beta(C)$

$$\Lambda(C) = \Lambda(\beta(C)) \cap \left(V_F \setminus V_F^C\right). \tag{4}$$

**Definition 1.4.** (Cluster collisions) Two or more clusters $\{C_i\}$ *collide* due to a set of fault nodes $\Delta_F$ if

$$\Delta_F \subseteq \bigcup_i \Lambda(C_i) \text{ and } \bigcap_i \beta(C_i) \cap \Lambda(\Delta_F) \neq \emptyset. \tag{5}$$

The LSD algorithm takes as input the matrix $H \in \mathbb{F}_2^{m \times n}$, where $m = |V_D|$, $n = |V_F|$, a syndrome $s \in \mathbb{F}_2^m$, and a reliability vector that contains the soft information $\lambda \in \mathbb{R}^n$. In the following, we will assume that $\lambda$ takes the form of *log-likelihood-ratios* (LLRs) such that the lower the LLR, the higher the probability that the corresponding fault belongs to the error. For instance, this is the form of soft information that is returned by the BP decoder.

A sequential version of the algorithm is outlined below and detailed in the pseudo-code in Box 1. A parallel version of the LSD algorithm is presented in Section 2 of the Supplementary Material

1. A cluster is created for each flipped detector node $d_i$ where $s_i = 1$. This cluster is represented by its corresponding sub-matrix $H_{[C_i]}$. Initially, each cluster is added to a list of invalid clusters.

2. Every cluster is grown by a single node $v_j$ drawn from the list of candidate nodes $\Lambda(C_i)$. For the first growth step after cluster initialization, we define $\Lambda(C_i) = \Lambda(\{s_i\})$ – see Definition IV.3. The chosen growth node $v_j \in \Lambda(C_i)$ in each step is the fault node with the highest probability of being in error. That is, $v_j$ has the lowest value among the LLRs for the candidate fault nodes $\lambda_j < \lambda_{j+1} < \cdots < \lambda_\ell$, $\ell = |\Lambda(C)|$. Hence, the growth step involves adding one new column to the cluster matrix $H_{[C_i]}$.

3. During growth, the algorithm detects collisions between clusters due to the selected fault nodes. Clusters that collide are merged.

4. The Gaussian elimination row operations performed on previous columns are performed on the new column together with the row operations needed to eliminate the newly added columns of $H_{[C_i]}$. In addition, every row operation applied to $H_{[C_i]}$ is also applied to the local syndrome $s_{[C_i]}$. This allows the algorithm to efficiently track when the cluster becomes valid. Explicitly, the cluster is valid when the syndrome becomes linearly dependent on the cluster decoding matrix i.e., when $s_{[C_i]} \in \text{image}(H_{[C_i]})$. In addition to cluster validation, the Gaussian elimination at each step enables an on-the-fly computation of the PLU factorization of the local cluster. We refer the reader to subsection "On-the-fly elimination" for an outline of our method.

5. The valid clusters are removed from the invalid cluster list, and the algorithm continues iteratively until the invalid cluster list is empty.

6. Once all clusters are valid, the local solutions $\hat{e}_{[C_i]}$ such that $H_{[C_i]} \cdot \hat{e}_{[C_i]} = s_{[C_i]}$ can be computed via the PLU decomposition of each cluster matrix $H_{[C_i]}$ that has been computed on-the-fly during cluster growth. The output of the LSD algorithm is the union of all the local decoding vectors $\hat{e}_{[C_i]}$.

### On-the-fly elimination
A common method for solving linear systems of equations is to use a matrix factorization technique. A foundational theorem in linear algebra states that every invertible matrix $A$ factorizes as $A = PLU$, that is, there exist matrices $P$, $L$, $U$ such that

$$A = PLU, \tag{6}$$

## BOX 1

# Localized statistics decoding (LSD) – serial algorithm

```
 1  H: decoding matrix
 2  s: syndrome vector
 3  λ: fault node soft information vector
 4  𝕀 := []: list of invalid clusters cl_i
 5  𝕍 := []: list of valid clusters cl_ℓ
 6  for s_i ∈ s do
 7  │   cl_i = CREATE_CLUSTER(s_i)
 8  │   𝕀.add(cl_i)
 9  while 𝕀 ≠ [] do
10  │   for cl ∈ 𝕀 do
11  │   │   cl.GROW_CLUSTER(λ)
12  │   for cl_i, cl_ℓ ∈ 𝕀 ∪ 𝕍 do
            // check if any clusters cl_i, cl_ℓ collide
13  │   │   merged = CHECK_COLLISION(cl_i, cl_ℓ)
14  │   │   if merged then
15  │   │   │   cl_{i∪ℓ} = MERGE_CLUSTERS(cl_i, cl_ℓ)
16  │   │   │   𝕀.remove({cl_i, cl_ℓ})
17  │   │   │   𝕍.remove({cl_i, cl_ℓ})
18  │   │   │   𝕀.add(cl_{i∪ℓ})
19  │   for cl ∈ 𝕀 do
20  │   │   cl.PLU_DECOMPOSE()
21  │   │   valid = cl.CHECK_VALIDITY(s_[cl])
22  │   │   if valid then
23  │   │   │   𝕍.add(cl)
24  │   │   │   𝕀.remove(cl)
25  local_decodings = []
26  for cl ∈ 𝕍 do
27  │   local_decodings.append(cl.PLU_SOLVE(s_[cl]))
28  return GLOBAL_DECODING(local_decodings)
```

where $P$ is a permutation matrix, $U$ is upper triangular, and $L$ is lower triangular with 1 entries on the diagonal. Once in PLU form, a solution $x$ for the system $A \cdot x = y$ can be efficiently computed using the forward and back substitution procedure[46]. The computational bottleneck of this method to solve linear systems stems from the Gaussian elimination procedure required to transform $A$ into PLU form.

Here, we present a novel algorithm called *on-the-fly elimination* to efficiently compute the PLU factorization over $\mathbb{F}_2$. Note that the algorithm can in principle be generalized to matrices over any field. However, in the context of coding theory, $\mathbb{F}_2$ is most relevant and we restrict the discussion to this case.

The main idea of the on-the-fly elimination is that row operations can be applied in a column-by-column fashion. If the operations that have been applied to each column of the matrix are stored, they can be applied to a newly added column such that only this column needs to be eliminated as all other columns are already in reduced form. This highlights the nice interplay between cluster growth (i.e., appending columns) and the on-the-fly elimination for PLU factorization of the cluster matrix.

To grow and merge clusters, multiple smaller steps are necessary. As detailed above, these steps include identifying fault nodes/column indices of the decoding matrix $H$ by which the invalid clusters will grow and determining whether an added fault node will lead to two or more clusters merging into a single one – see Definition IV.4. For simplicity, we first describe the case of sequential cluster growth. Our on-the-fly procedure can analogously be applied in a parallel implementation, see Section 2 of the Supplementary Material.

Let $C_i$ be an active cluster, that is, $(H_{[C_i]}, s_{[C_i]})$ does not define a solvable decoding problem as $s_{[C_i]} \notin \text{image}(H_{[C_i]})$. To grow cluster $C_i$, we consider candidate fault nodes $v_j \in \Lambda(C_i)$ – fault nodes not already in $C_i$ but connected to check nodes on its boundary $\beta(C)$, see Definition IV.3. The candidate fault node with the highest probability of being in error according to the soft information vector $\boldsymbol{\lambda} \in \mathbb{R}^n$ is selected. Once $v_j$ has been chosen, we check whether its neighboring detector nodes are boundary nodes of any other (valid or invalid) clusters i.e., we check for collisions, cf Definition IV.4. If this is not the case, we proceed as follows. We now assume that the active cluster $C_i$ described by sub-matrix $H_{[C_i]}$

that has a PLU factorization of the form

$$H_{[C_i]} = P_i L_i U_i, \tag{7}$$

where $P_i$, $L_i$, $U_i$ are as in Eq. (6). Adding a fault node $v_j$ to the cluster is equivalent to adding a (sparse) column vector $\mathbf{b}$ to $H_{[C_i]}$, i.e.,

$$H_{[C_i \cup \{v_j\}]} = \left( \begin{array}{c|c} H_{[C_i]} & \mathbf{b} \\ 0 & \end{array} \right). \tag{8}$$

A key insight is that the PLU factorization of the extended matrix $H_{[C_i \cup \{v_j\}]}$ can be computed through row operation on column $\mathbf{b}$ alone: it is not necessary to factorize the full matrix $H_{[C_i \cup \{v_j\}]}$ from scratch. By applying the PLU factorization of $H_{[C_i]}$ block-wise to the extended matrix $H_{[C_i \cup \{v_j\}]}$, we obtain

$$\left( \begin{array}{c|c} U_i & L_i^{-1} P_i^T \mathbf{b}_i \\ 0 & \mathbf{b}_{i*} \end{array} \right), \tag{9}$$

where $\mathbf{b}_i$ is the projection of $\mathbf{b}$ onto the detectors/row coordinates that are enclosed by $C_i$. Similarly, $\mathbf{b}_{i*}$ is the projection onto detector coordinates not enclosed by $C_i$. Importantly, applying the operators $L_i^{-1}$ and $P_i^T$ does not affect the support of $\mathbf{b}_i$ and $\mathbf{b}_{i*}$, as both these operators act solely on the support of $C_i$. Combining this with Eq. 9, we note that to complete the PLU factorization of $H_{[C_i \cup \{v_j\}]}$ only $\mathbf{b}_{i*}$ has to be reduced, which has a computational cost proportional to its weight – crucially only a small constant for bounded LDPC matrices $H$.

We now continue by describing the collision case, where the addition of a fault node to a cluster results in the merging of two clusters. The generalization to the merging of more than two clusters is straightforward.

Suppose that the selected fault node $v_j$ by which the cluster $C_i$ is grown is connected to a check node in the boundary $\beta(C_\ell)$, with $C_i \neq C_\ell$. Let $\mathbf{b}$ be the column of $H$ associated with the fault node $v_j$. Re-ordering its coordinates if necessary, we can write $\mathbf{b}$ as $(\mathbf{b}_i, \mathbf{b}_\ell, \mathbf{b}_*)$ where $\mathbf{b}_i$, $\mathbf{b}_\ell$, and $\mathbf{b}_*$ are the projections of $\mathbf{b}$ on the row coordinates contained in $C_i$, $C_\ell$, and neither of them, respectively. Thus, using a block matrix notation, for the combined cluster $C_i \cup C_\ell \cup \{v_j\}$, we have

$$H_{[C_i \cup C_\ell \cup \{v_j\}]} = \left( \begin{array}{c|c|c} H_{[C_i]} & 0 & \mathbf{b}_i \\ \hline 0 & H_{[C_\ell]} & \mathbf{b}_\ell \\ \hline 0 & 0 & \mathbf{b}_* \end{array} \right). \tag{10}$$

By applying the PLU factorization of $H_{[C_i]}$ and $H_{[C_\ell]}$ block wise, we can put $H_{[C_i \cup C_\ell \cup \{v_j\}]}$ into the form

$$\left( \begin{array}{c|c|c} U_i & 0 & L_i^{-1} P_i^T \mathbf{b}_i \\ \hline 0 & U_\ell & L_\ell^{-1} P_\ell^T \mathbf{b}_\ell \\ \hline 0 & 0 & \mathbf{b}_* \end{array} \right). \tag{11}$$

Since $U_i$ and $U_\ell$ are, in general, not full rank, they may contain some zero rows. As a result, the first $|C_i| + |C_\ell|$ columns are not necessarily in reduced form. To make this issue clearer, we introduce the notation $\mathbf{u}_m = L_m^{-1} P_m \mathbf{b}_m$ for $m \in \{i, \ell\}$ and express the above matrix as

$$\left( \begin{array}{c|c|c} U_i & 0 & \mathbf{u}_i \\ \hline 0 & U_\ell & \mathbf{u}_\ell \\ \hline 0 & 0 & \mathbf{b}_* \end{array} \right) = \left( \begin{array}{c|c|c} U_{i,\bullet} & 0 & \mathbf{u}_{i,\bullet} \\ \hline 0 & 0 & \mathbf{u}_{i,\perp} \\ \hline 0 & U_{\ell,\bullet} & \mathbf{u}_{\ell,\bullet} \\ \hline 0 & 0 & \mathbf{u}_{\ell,\perp} \\ \hline 0 & 0 & \mathbf{b}_* \end{array} \right). \tag{12}$$

Here, by slight misuse of notation, we group the non-zero rows of $U_m$ in the index set $(m, \bullet)$, and its zero rows in the set $(m, \perp)$; we regroup the coordinates of vector $\mathbf{u}$ accordingly. We remark that the row sets $\bullet$ and $\perp$ are distinct from the row set $*$ identified when writing $\mathbf{b}$ as the combination of its projection onto row coordinates enclosed by $C_i$ and outside it. By identifying the appropriate row sets for the clusters $C_i$, $C_\ell$ and the fault node $\{v_j\}$ as detailed in Eq. (12), we can construct a block-swap matrix to bring $H_{[C_i \cup C_\ell \cup \{v_j\}]}$ into the form

$$\left( \begin{array}{c|c|c} U_{i,\bullet} & 0 & \mathbf{u}_{i,\bullet} \\ \hline 0 & U_{\ell,\bullet} & \mathbf{u}_{\ell,\bullet} \\ \hline 0 & 0 & \mathbf{u}_{i,\perp} \\ \hline 0 & 0 & \mathbf{u}_{\ell,\perp} \\ \hline 0 & 0 & \mathbf{b}_* \end{array} \right), \tag{13}$$

and similarly for its PLU factors. Since the matrix in Eq. 13 has the same form as the one in Eq. 8, the algorithm can proceed from this point onward as in the case of the addition of a single fault node to a cluster. In conclusion, via a swap transformation, we can effectively reduce the problem of merging two clusters to the problem of adding a single fault node to one cluster.

## Numerical decoding simulations

For all numerical simulations in this work, we employ a circuit-level noise model that is characterized by a single parameter $p$, the physical error probability. Typically, the standard noise model for each time step is then to assume the following.

- Idle qubits are subject to depolarizing errors with probability $p$.
- Pairs of qubits acted on by two-qubit gates such as CNOT are subject to two-qubit depolarizing errors *after* the gate, that is, any of the 15 non-trivial Pauli operators occurs with probability $p/15$.
- Qubits initialized in $|0\rangle(|+\rangle)$ are flipped to $|1\rangle$ $(|-\rangle)$ with probability $p$.
- The measurement result of an $X/Z$ basis measurement is flipped with probability $p$.

For surface code simulations, we use the syndrome extraction circuits and noise model provided by Stim[30]. We note that this noise model is similar to the one described above, however, it differs in small details such as that it combines measurement and initialization errors, ignores idling errors and applies a depolarizing channel to data qubits prior to each syndrome measurement cycle. We perform a memory experiment for a single check side (Z-checks), called surface_code:rotated_memory_z experiment in Stim, over $d$ syndrome extraction cycles for code instances with distance $d$.

The syndrome extraction circuits for the family of HGP codes presented in Section 3 of the Supplementary Material and results presented in subsection "Numerical results" are obtained from the minimum edge coloration of the Tanner graphs associated to the respective parity check matrix, see ref. 13 for details. In particular, we generate associated Stim files of $r = 12$ noisy syndrome extractions using a publicly available implementation of the aforementioned coloration circuit by Pattison[57]. In this case, the standard circuit-level noise model described at the beginning of this section is employed. We decode $X$ and $Z$ detectors separately using a $(3, 1)$ − overlapping window decoder. That is, for each decoding round, the decoder obtains the detection events for $w = 3$ syndrome extraction cycles and computes a correction for the entire window. However, it only applies the correction for a single $(c = 1)$ syndrome extraction cycle, specifically the one that occurred the furthest in the past. For more details on circuit-level overlapping window decoding, see ref. 58. We have chosen $w = 3$ as this was the value used in ref. 13. Note that it is possible that

(small) decoding improvements could be observed by considering larger values of ($w$, $c$) for the overlapping window decoder.

The BB codes are simulated using the syndrome extraction circuits specified in ref. 12, and the `Stim` files are generated using the code in ref. 59. There, the authors recreate the circuit-level noise model described in ref. 12 which, up to minor details, implements the noise model described at the beginning of this section. Similar to the HGP codes mentioned above, we decode $X$ and $Z$ decoders separately. Analogous to our surface code experiments, we simulate for a distance $d$ code $d$ rounds of syndrome extraction and decode the full syndrome history at once. As the BB codes are CSS codes, we decode $X$ and $Z$ detectors separately.

If not specified otherwise, we have used the min-sum algorithm for BP, allowing for a maximum of 30 iterations with a scaling factor of $\alpha = 0.625$, using the parallel update schedule. We have not optimized these parameters and believe that an improved decoding performance, in terms of speed and (or) accuracy, can be achieved by further tweaking these parameters.

## Parallel algorithm

We propose a parallel version of the LSD algorithm (P-LSD) in Section 2 of the Supplementary Material that uses a parallel data structure, inspired by refs. 60,61, to minimize synchronization bottlenecks. We discuss the parallel algorithm in more detail in Section 2 of the Supplementary Material. There, we derive a bound on the parallel *depth* of P-LSD, that is, roughly the maximum overhead per parallel resource of the algorithm. We show that the depth is $O(\text{polylog}(n) + \kappa^3)$ in the worst-case, where $n$ is the number of vertices of the decoding graph and $\kappa$ is the maximum cluster size. A crucial factor in the runtime overhead of P-LSD is given by the merge and factorization operations. We contain this overhead by (i) using the parallel union-find data structure of ref. 60 for cluster tracking and (ii) using parallel on-the-fly elimination to factorize the associated matrices. If we assume sufficient parallel resources, the overall runtime of parallel LSD is dominated by the complexity of computing the decoding solution for the largest cluster. To estimate the expected overhead induced by the cluster sizes concretely, we (i) investigate analytical bounds and (ii) conduct numerical experiments to analyze the statistical distribution of clusters for several code families, see Section 1 of the Supplementary Material.

## Data availability

The simulation data generated in this study has been deposited in the Zenodo database and is available under[62].

## Code availability

The proposed algorithm and scripts to run the numerical experiments to generate the results presented above is publicly available on Github[42].

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

## Acknowledgements

The authors would like to thank C.A. Pattison for providing the code to generate the coloration circuits for the HGP code family via Github. T.H. acknowledges the financial support from the Chalmers Excellence Initiative Nano and the Knut and Alice Wallenberg Foundation through the Wallenberg Centre for Quantum Technology (WACQT). This work was done in part while L.B. was visiting the Simons Institute for the Theory of Computing. L.B. and R.W. acknowledge funding from the European Research Council (ERC) under the European Union's Horizon 2020 research and innovation program (grant agreement No. 101001318) and Millenion, grant agreement No. 101114305). This work was part of the Munich Quantum Valley, which is supported by the Bavarian state government with funds from the Hightech Agenda Bayern Plus, and has been supported by the BMWK on the basis of a decision by the German Bundestag through project QuaST, as well as by the BMK, BMDW, and the State of Upper Austria in the frame of the COMET program (managed by the FFG). J. R. is funded by an EPSRC Quantum Career Acceleration Fellowship (grant code: UKRI1224). J. R. further acknowledges support from EPSRC grants EP/T001062/1 and EP/X026167/1. The Berlin team has also been funded by BMBF (RealistiQ, QSolid), the DFG (CRC 183), the Munich Quantum Valley, the Einstein Research Unit on Quantum Devices, the Quantum Flagship (PasQuans2, Millenion), and the European Research Council (ERC DebuQC). For Millenion and the Munich Quantum Valley, this work is the result of joint-node collaboration.

## Author contributions

T.H., L.B., and J.R. implemented and conceived the LSD algorithm. T.H. performed the numerical decoding simulations and implemented the overlapping window decoder. A.Q. performed the numerical cluster simulations and formulated the cluster bounds. T.H., L.B., A.Q., J.E., R.W. and J.R. drafted the manuscript and contributed to analytical considerations.

## Funding

## Competing interests

The authors declare no competing interests.
