## [Transparent Peer Review file · Nature Communications]

Localized statistics decoding: A parallel decoding algorithm for quantum low-density parity-check codes

Corresponding Author: Mr Timo Hillmann

Version 0:

Reviewer comments:

Reviewer #4

(Remarks to the Author)

The authors have addressed many of the comments in their response, although there are still further changes that would be very desirable before publication.

The authors have provided very nice timing data and analysis in their response but this is not included in the manuscript or supplementary material - why is this? In my opinion a timing analysis should be presented in the main text, and at the very least in the supplement. The most significant impact of the paper is its improvement to practical running time and the open-source implementation, so presenting this analysis is very relevant to the key message of the paper.

The first figure in the reply that the authors present on running time ("Response to reviewer 2 Comment 2") is post-selected on the minority of cases where BP fails. I think the non-post-selected runtime ("Response to Reviewer 2 Comment 9") is more important as it shows the expected running time of the overall decoder, although this plot is also nice as a supplemental figure.

The figures in "Response to Reviewer 2 Comment 9" are excellent and in my view should be included in the main text. It would be nice if there was also an analysis at a higher error rate than $2e-3$ (e.g. $5e-3$ or $1e-2$). Although I agree with the authors that the below-threshold regime is more important, it is still good to quantify the disadvantages of the decoder at higher error rates - this could be added into the supplement.

In "Response to Reviewer 2 Comment 6" I suggest changing "leads to efficient factorizations" to "can lead to efficient factorizations" given the lack of a formal guarantee.

In "Response to Reviewer 2 Comment 8" I suggest changing "cannot be improved if" to "cannot be made lower weight" or similar, to be more precise.

I think the "Response to Reviewer 2 Comment 14" misses the point I am making. Surely in every instance (not just the worst case), the depth of the parallel decoder is still set by the size of the largest cluster in the instance. For any given instance, you still have to wait for the validity check of the largest cluster in the instance. Therefore it seems to me that the expected parallel runtime is set by the size of the largest cluster size not the average cluster size.

I am satisfied that the authors have otherwise addressed the comments made. I do agree with Reviewer 3 that it would be more suitable for it to be published in a more specialised journal such as npj Quantum Information although I can see that there is a case to be made for Nature Communications. Regardless of the journal, it is ready for publication after the remaining revisions are addressed.

(Remarks on code availability)

I have checked that the code installs and runs.

Reply to Reviewer 4

Reviewer 4 Comment 1

The authors have addressed many of the comments in their response, although there are still further changes that would be very desirable before publication.

Response to Reviewer 4 Comment 1:

We would like to thank the reviewer very much for the in-depth feedback and new comments on our manuscript.

Reviewer 4 Comment 2

The authors have provided very nice timing data and analysis in their response but this is not included in the manuscript or supplementary material - why is this? In my opinion a timing analysis should be presented in the main text, and at the very least in the supplement. The most significant impact of the paper is its improvement to practical running time and the open-source implementation, so presenting this analysis is very relevant to the key message of the paper.

Response to Reviewer 4 Comment 2:

We thank the reviewer for highlighting the importance of the timing data. We agree that the reduced runtime represents a key practical contribution of our work. In response to Comments 2, 3, and 4, we have revised both the main manuscript and the supplementary material accordingly.

Specifically, we have added a new section in the manuscript dedicated to our serial prototype of BP+LSD, which already demonstrates substantial improvements in runtime and feasibility compared to state-of-the-art BP+OSD implementations. The timing data previously provided, along with results from new experiments at higher error rates closer to the threshold (as noted in Comment 4), are now included in the supplementary material.

Overall, these results provide empirical support for our claim that even our open-source implementation achieves runtime advantages over BP+OSD, primarily due to the on-the-fly solving approach, which is particularly relevant for current numerical studies.

Runtime statistics.

To estimate the time overhead of the proposed decoder in numerical simulation scenarios and to compare it with a state-of-the-art implementation of BP+OSD, we present preliminary timing results for our prototypical open-source implementation of LSD in Section 3.4 of the

Supplementary Material. We note that for a more complete assessment of performance, it will be necessary to benchmark a fully parallel implementation of the algorithm, designed for specialized hardware such as GPUs or FPGAs. We leave this as a topic for future work.

Reviewer 4 Comment 3

The first figure in the reply that the authors present on running time ("Response to reviewer 2 Comment 2") is post-selected on the minority of cases where BP fails. I think the non-post-selected runtime ("Response to Reviewer 2 Comment 9") is more important as it shows the expected running time of the overall decoder, although this plot is also nice as a supplemental figure.

Response to Reviewer 1 Comment 3:

Once again, we thank the reviewer for this valuable observation. We agree that the non-post-selected runtime data (presented in our response to Reviewer 2 Comment 9) offers a more accurate estimate of the expected runtime of the complete decoder and is therefore more relevant for assessing its practical performance.

In the revised supplementary material, we now include both the non-post-selected and post-selected timing plots. The former provides a realistic picture of the overall decoder runtime, while the latter isolates the cost of the post-processing subroutine. This distinction is useful for evaluating the scalability and impact of replacing OSD post-processing with LSD post-processing.

Reviewer 4 Comment 4

The figures in "Response to Reviewer 2 Comment 9" are excellent and in my view should be included in the main text. It would be nice if there was also an analysis at a higher error rate than $2e-3$ (e.g. $5e-3$ or $1e-2$). Although I agree with the authors that the below-threshold regime is more important, it is still good to quantify the disadvantages of the decoder at higher error rates - this could be added into the supplement.

Response to Reviewer 1 Comment 4:

We thank the referee for suggesting to include these additional timing statistics. We have added non-postselected timing statistics for the gross code at $3e-3$, $5e-3$, and $7e-3$ physical error rate in the supplementary material, see below.. At these error rates, especially $3e-3$ and $5e-3$, a good separation of the time-scales of BP and the LSD routines are apparent.

Reviewer 4 Comment 5

In "Response to Reviewer 2 Comment 6" I suggest changing "leads to efficient factorizations" to "can lead to efficient factorizations" given the lack of a formal guarantee.

Response to Reviewer 1 Comment 5:

We agree with the reviewer that the formulation should be a little more careful, as indicated. Hence, we have adapted the sentence to:

This weighted growth strategy is crucial for controlling the cluster size: limiting growth to a single fault node per time step increases the likelihood that an efficient factorization is found, especially for QLDPC codes with high degrees of expansion in their decoding graphs.

Reviewer 4 Comment 6

In "Response to Reviewer 2 Comment 8" I suggest changing "cannot be improved if" to "cannot be made lower weight" or similar, to be more precise.

Response to Reviewer 1 Comment 6: We thank the reviewer for this suggestion. We agree that "cannot be made lower weight" is a more precise formulation and have updated the phrasing accordingly in the revised manuscript.

Reviewer 4 Comment 7

I think the "Response to Reviewer 2 Comment 14" misses the point I am making. Surely in every instance (not just the worst case), the depth of the parallel decoder is still set by the size of the largest cluster in the instance. For any given instance, you still have to wait for the validity check of the largest cluster in the instance. Therefore it seems to me that the expected parallel runtime is set by the size of the largest cluster size not the average cluster size.

Response to Reviewer 1 Comment 7:

We thank the reviewer for their comment. They are right that the parallel runtime for any given shot is determined by the size of the largest cluster in that instance. Therefore, the expected parallel runtime is in principle governed by the expected value of this maximum, $E(\kappa)$, rather than the average cluster size κ_α .

Our intention has not been to contradict this, but rather to argue, based on numerical evidence, that the distribution of cluster sizes is typically narrow, with small variance across shots. In such cases, the average cluster size κ_α provides a reasonable and practical *proxy* for the expected maximum cluster size $E(\kappa)$, particularly when aiming to capture broad trends or compare decoder behavior under different codes. We have now clarified this.

Furthermore, it is important to note that the number of growth steps required to reach a cluster of size κ is not necessarily equal to κ due to cluster merges. Indeed, cluster merges must occur (almost always) for clusters to become valid, which complicates a direct interpretation of cluster size and growth time. This additional complexity supports the use of κ_α as a useful statistical figure of merit in practice.

Reviewer 4 Comment 8

I am satisfied that the authors have otherwise addressed the comments made. I do agree with Reviewer 3 that it would be more suitable for it to be published in a more specialised journal such as npj Quantum Information although I can see that there is a case to be made for Nature Communications. Regardless of the journal, it is ready for publication after the remaining revisions are addressed.

Response to Reviewer 1 Comment 8:

We thank the reviewer for their positive response and for proposing that an updated version of the manuscript is ready for publication. We have carefully addressed all concerns and have given a point by point reply to ensure that all aspects are addressed accordingly.

Reviewer 4 Comment 9 (Remarks on code availability):

I have checked that the code installs and runs.

Response to Reviewer 1 Comment 9:

We thank the reviewer for appreciating the open-source code and confirming that it works as expected.